# Feasibility of Screening Programs for Domestic Violence in Pediatric and Child and Adolescent Mental Health Services: A Literature Review

**DOI:** 10.3390/brainsci12091235

**Published:** 2022-09-13

**Authors:** Elena Arigliani, Miriam Aricò, Gioia Cavalli, Franca Aceti, Carla Sogos, Maria Romani, Mauro Ferrara

**Affiliations:** Section of Child and Adolescence Neuropsychiatry, Department of Human Neurosciences, Sapienza University of Rome, Via dei Sabelli 108, 00185 Rome, Italy

**Keywords:** domestic violence, pediatric care, mental health, child development, screening, abused children, aggressive behaviors

## Abstract

Each year, 275 million children worldwide are exposed to domestic violence (DV) and suffer negative mental and physical health consequences; however, only a small proportion receive assistance. Pediatricians and child psychiatrists can play a central role in identifying threatened children. We reviewed experiences of DV screening in pediatric and child and adolescent mental health services (CAMHS) to understand its feasibility and provide clues for its implementation. We performed bibliographic research using the Sapienza Library System, PubMed, and the following databases: MEDLINE, American Psychological Association PsycArticles, American Psychological Association PsycInfo, ScienceDirect, and Scopus. We considered a 20-year interval when selecting the articles and we included studies published in English between January 2000 and March 2021. A total of 23 out of 2335 studies satisfied the inclusion criteria. We found that the prevalence of disclosed DV ranged from 4.2% to 48%, with most prevalence estimates between 10% and 20%. Disclosure increases with a detection plan, which is mostly welcomed by mothers (70–80% acceptance rates). Written tools were used in 55% of studies, oral interviews in 40%, and computer instruments in 20%. Mixed forms were used in three studies (15%). The most used and effective tool appeared to be the Conflict Tactics Scale (CTS) (30% of studies). For young children, parental reports are advisable and written instruments are the first preference; interviews can be conducted with older children. Our research pointed out that the current literature does not provide practical clinical clues on facilitating the disclosure in pediatric clinics and CAMHS. Further studies are needed on the inpatient population and in the field of children psychiatry.

## 1. Introduction

The terms intimate partner violence (IPV) and domestic violence (DV) are often used to define the same phenomenon, referring to both acts and threats of physical, sexual, psychological and emotional violence, perpetrated by a current or former partner or spouse [1]. Usually, the concept of DV is related to IPV, especially in the American and North Europe contexts, but sometimes, it includes all types of violence that occur within families [2].

DV is a common phenomenon; according to the report by the World Health Organization (WHO) in 2013, data on women report a lifetime DV prevalence of 30% worldwide [3], with a higher risk of DV exposure in pregnant mothers and younger children [4]. Regarding the American population, the number of victims of DV is estimated to be 15.5 million [5]. IPV and DV are currently recognized as forms of child abuse (CA) [6]. Despite not being direct victims of violence, children suffer lifelong adverse consequences from growing up in a harsh environment [7]. In particular, they tend to exhibit more psychosocial problems by internalizing and externalizing concerns [8]. Children of violent couples are also highly likely to be victims of other types of violence (“double whammy” phenomenon) [9] and are more prone to act violently in extra-domestic settings, leading to the so-called “cycle of violence” [10].

The identified cases of children exposed to DV are still minimal and only a small proportion receive assistance from child protection services [11], despite widespread awareness of the clinical consequences of exposure to violence. A gap seems to exist between the entity of the phenomenon, in terms of prevalence and morbidity, and the ability of the healthcare system to identify exposed children.

It is becoming increasingly clear that children who are direct victims of DV can be the protagonist of screening programs [12]. Indeed, the first opinions on the theme date back to early 1990, when some authors underlined the need for interviews regarding IPV in clinical settings [13]. In 2010, the American Academy of Pediatricians released a clinical report on children’s exposure to IPV and the role of pediatricians [14], which followed a previous position statement, highlighting the active role of health care providers in intercepting high-risk situations regarding IPV through appropriate information and training for intervention [15]. In the setting of primary care of children, most of the implemented IPV screening programs have been addressed to women rather than children, but there is still little evidence about performing universal versus risk-based screening [14]. Moreover, detailed guidelines for clinicians on how to perform the screening are still missing [16]. While pediatricians have been dealing with the idea of IPV screening for at least a decade, children’s psychiatrists do not seem to have addressed the issue, although the most immediate consequence on children’s health is related to psychopathological problems [2].

The present review addresses the need to revise the current literature in order to make clear the state of the art on screening for DV in pediatric clinics and in child and adolescent mental health services (CAMHS). Indeed, clinicians still lack awareness and practical guidelines on the theme, and our study aims to point out any valuable clinical clues that can be helpful in children’s care settings. Moreover, our research aims to answer the following questions: in the pediatric population, does screening for DV increase the likelihood of detecting exposed children? Which is the best place, among pediatric health-care services, where screening should take place? Which patients should the screening involve? Which instruments are recommended, and which cautions are needed?

## 2. Material and Methods

### 2.1. Protocol

The methods of analysis and inclusion criteria were predefined and registered according to an internal study protocol.

### 2.2. Eligibility Criteria

We included studies published in English between January 2000 and March 2021 on children under 18 years of age or on their caregivers in pediatric or mental health clinics that investigated screening strategies for DV exposure. Studies were excluded if they were any of the following: (i) systematic reviews; (ii) letters to editors; (iii) single case reports; or (iv) studies generally referring to adverse childhood experiences (ACEs), child maltreatment, child trauma, or CA (Table 1).

### 2.3. Sources of Information

The selected studies were identified through bibliographic research using electronic databases. Only articles in English were considered. Bibliographic research was conducted using the Sapienza Library System (Sistema Bibliotecario Sapienza, SBS), PubMed, and the following databases: MEDLINE (1966–present), American Psychological Association (APA) PsycArticles (1894–present), APA PsycInfo (1967–present), ScienceDirect, and Scopus. We considered a 20-year interval when selecting the articles. The last literature search was performed on 2 March 2021.

The following keywords were used for this search: domestic violence, family violence, and intimate partner violence, combined either with the word pediatric or the words children AND adolescent AND psychiatric clinic.

### 2.4. Study Selection

Eligibility assessment was performed through standardized open processes conducted by two independent reviewers. Any disagreement between the two reviewers was resolved by consensus. The first screening was performed by one reviewer, who reviewed the titles and abstracts. The selected articles then underwent full-text evaluation by the two reviewers independently in order to identify the most relevant studies according to our eligibility criteria.

### 2.5. Data Collection Process

Data were extracted according to the above-mentioned objective by two investigators independently; data were cross-verified for accuracy and completeness. Extracted data included the following: setting where the study was held, population characteristics, prevalence rates of DV, characteristics of the exposed sample, instruments used for the screening process, their methods of administration and their acceptability. Two reviewers independently assessed the methodological quality of the included studies.

### 2.6. Data Synthesis and Analysis

We described the results of the studies qualitatively and used tables of evidence. Studies were grouped according to the target population of the screening assessment (either caregivers or children).

## 3. Results

### 3.1. Available Literature

The first literature search yielded 2335 articles. After rejection of duplicates, 1742 titles and abstracts were read and 1622 were excluded based on the exclusion criteria. We then examined the remaining 120 full-text articles. Six additional articles were added through a backward reference search and evaluated for eligibility. A total of 23 papers were selected according to eligibility criteria (Table 1). The flowchart summarizes the selection process used for the present review (Figure 1).

The settings of all studies are shown in Table 2. There were considerably more studies on the topic of DV screening in pediatric clinics vs. CAMHS (18 vs. 5; 78% vs. 22%). We found more studies in primary care clinics [12,17,18,19,20,21,22,23,24,25,26] (11, 48%) than in pediatric emergency departments (EDs) (5, 22%) [27,28,29,30,31]. Only one study included data on inpatients [31]. Twenty articles (87%) focused on DV screening in mothers [12,17,18,19,20,21,22,23,24,25,26,27,28,29,30,31,32,33,34,35], while three (13%) focused on DV screening in children and adolescents [2,36,37]. We will discuss these studies separately.

### 3.2. Studies on DV Screening in Mothers

#### 3.2.1. Settings and Samples

Studies were heterogeneous in terms of sample size and timeframe considered for exposure (Table 3). Most studies used a convenience sample recruited from clinical practices or EDs. The sample size varied widely, from only 90 families [34] to over 13,000 units [29]. In all studies except one [34], only mothers were selected for screening and the presence of a male partner was an exclusion criterion to guarantee the woman’s safety [23,25,26,27,30,32,38]. In many cases, if a child older than 3 years of age was present, the interview was not performed unless it was possible to send the child out of the room [21], an important limitation in pediatric clinics, where separating children from parents might not be feasible [39]. Demographic characteristics of the total sample were often not specified. Overall, studies were conducted in areas with an adequate variety of sociodemographic characteristics; the population ranged from married couples mostly covered by private insurance [12] to disadvantaged communities, with rates of adherence to aid programs (e.g., Medicaid) up to 93% and rates of single parenthood up to 85% [17]. Most studies reached women between 20 and 30 years of age who were mothers of young children, mostly preschoolers.

#### 3.2.2. Disclosure Rates

The prevalence of disclosed DV ranged from 4.2% [31] to 48% [34], with most prevalence estimates between 10% and 20%. Data from CAMHS were fewer and more heterogeneous; a prevalence of 48% was found with a very small sample [34], while another showed a prevalence of 21% [35]. When DV that occurred in the past 12 months was investigated, the prevalence was found to be lower in some studies (between 0.5% [24] and 3.7% [21]), but not in others (10% [38]–11% [27]). When current abuse was investigated, the prevalence decreased drastically, ranging from 2% to 2.5% [12,22]. However, no consistent definitions of “current” or “past abuse” [21,28,29] are available.

#### 3.2.3. Characteristics of Exposed Sample

Only eight studies (40%) provided specific data on women exposed to DV [12,18,20,26,27,32,34,35]. A higher risk of DV exists for young women [18], families more often involved in criminal investigations and/or parental custody battles [35], mothers who report a history of harm to the child [12], mothers who are in a relationship other than the first marriage [12], those with four or more children [12], those who are eligible for WIC (Special Supplemental Nutrition Program for Women, infants and Children), and those who previously “no-showed” for a child’s wellbeing visit [12]. No relation was found between DV and race, ethnicity, poverty, or the child’s diagnosis (illness or injury) [27], and no significant differences were found in terms of marital status, income, number of children [18], sex, children’s and mothers’ age or nationality [35]. One study showed a higher risk for women who had not completed high school [27], but two studies did not confirm this data [18,35].

Only three of the examined studies (15%) specifically explored the psychopathology of parents or children [18,19,34]; they reported higher frequency of depressive symptoms [18] in exposed mothers and more common behavioral [19], internalizing and externalizing problems [18,34] that increased with age [19].

#### 3.2.4. Screening Instruments and Acceptability

DV was directly assessed in all studies, since parental reporting is essential for disclosure [31]. The studies differed according to the instruments used and the method of administration (Table 4 and Table 5). Written tools were used in 55% of the studies, oral interviews in 40%, and computer instruments in 20%. Mixed forms were used in three studies (15%) [23,25,38]. The most used and effective tool appeared to be the Conflict Tactics Scale (CTS) [40] (30% of studies). The use of a shorter questionnaire as a first screening tool appeared advantageous [17,24], while the extended CTS provided a detailed characterization of the type of violence [24]. In contrast, the Partner Violence Screen (PVS) [41] was used for an initial assessment in 15% of the studies, especially those conducted in Eds. [21,27,29,38]. The PVS has the advantage of short and direct questions and the disadvantage of being limited to the previous 12 months [21]. Some new screening tools showed low sensitivity [17,23], although they might be sufficient to permit disclosure from women who are ready to disclose violence. Women showed good acceptance of DV screening within the clinical setting (70–80% acceptance rates [12,24,27]). No differences in DV disclosure rates between different formats, including verbal, written, computer [23], or audiotape questionnaires, were found in two studies, although a better outcome of oral surveys in comparison to written interviews was reported in one study. However, DV prevalence in the written interview group was noticeably low (0%) [25]. A caregiver-initiated computerized system had the advantage of recruiting large samples and was appreciated in terms of privacy, although it was not compared to any other method in the study [29]. Women preferred direct verbal questioning in the study of Newman et al. [27], and audiotapes in the study by Bair-Merritt et al. [38]. Findings regarding the use of informative materials to facilitate disclosure within the pediatric setting were controversial and inconclusive [28,33].

### 3.3. Studies on DV Screening in Children

#### 3.3.1. Setting and Samples

Three of the selected studies investigated the feasibility of screening for DV in minors recruited from CAMHS in three countries, Sweden [2], Austria [37], and Spain [36] (Table 6). The three studies together enrolled a total of 1891 children aged 6 to 20 years, with an equal gender distribution. Study designs were similar; they explored personal experiences of interpersonal violence using an interview addressed to children. The study by Olaya et al. investigated DV exclusively [36], while Hultmann and Broberg and Völkl-Kernstock et al. explored DV among other forms of interpersonal violence [2,37].

#### 3.3.2. Disclosure Rates

The study by Olaya et al. found a DV prevalence of 19.2% [36]. Hultmann et al. reported a prevalence of family violence (FV) of 48%. This figure included 21% who were victims of FV only (21%) and 27% who were exposed to poly-victimization (FV and interpersonal violence) [2]. A total of 67% of responders experienced some type of violence. Völkl-Kernstock et al. found 75% of the sample had experienced violence, with DV being the most frequently reported (27% of the total sample) [37].

#### 3.3.3. Characteristics of Exposed Sample

The most represented age group in the violence-exposed samples ranged between 11 and 17 years of age [2,36,37]. Regarding gender distribution, a lower percentage of males was found in the exposed group [36], especially in older ages [37], although males were prevalent in the group exposed to interpersonal violence [2]. Therefore, DV prevalence ranged from 19.2% to 48% and was the most frequent form of violence, being more common in females than males.

The exposed group presented a higher frequency of one-parent families [36], more frequent economic problems [36], and an increased probability of living with neither parent or under one-parent custody or being born abroad [2]. There were no significant differences between the two groups in terms of maternal and paternal age, education level, occupation, socioeconomic status, or perception of their neighborhood from a social point of view [36].

The study by Olaya et al. provided the most detailed information about parenting style as characterized by rejection, low emotional warmth, and less control [36]. Physical punishment was more frequent among fathers, whereas mothers were more overprotective towards males and more prone to physical punishment towards females [36]. Psychopathology was common within abusive families [36].

Psychiatric assessment of patients was a key point of all the selected studies. Self-administered questionnaires and clinical interviews were used. A higher number of diagnoses and symptoms in DV exposed children [36] and a significant correlation between DV and clinical diagnosis [37] were found. Exposed children showed a higher frequency of dysthymic disorder, posttraumatic stress disorder [36], adjustment disorder, and attention-deficit or disruptive behavior diagnoses [2]. Males were more likely to be diagnosed with an externalizing diagnosis, while females most often received an internalizing diagnosis [37]. We found higher scores on externalizing and rule-breaking behavior subscales of the Child Behavior Checklist (CBCL), higher daily global impairment and an increased risk of self-harming behaviors [36], and higher scores on the total problem scale and peer problems on the Strengths and Difficulties Questionnaire (SDQ) [2]. Global functioning was overall lower in children in the exposed group [2].

#### 3.3.4. Screening Instruments and Acceptability

The three selected studies approached DV detection from the child’s point of view.

Olaya et al. [36] used two items from the Children’s Perception of Interparental Conflict Scale (CPIS) [42] and Hultmann and Broberg [2] used a modified version of the Life Incidence of Traumatic Events (LITE) [43]. Children who responded positively to one of the LITE questions were considered in the FV group, so that in this study, the sample was actually a group of poly-victimized children [2]. The scale, in the form of a questionnaire, was administered to children (without parents in the room) by clinicians during the first visit [2]. Völkl-Kernstock et al. [37] used the Childhood Trauma Interview (CTI) [44]. The investigation was performed by three child psychiatry residents in the form of a semi-structured interview. The authors did not specify which items were used to rate a child as positive for DV, and when describing the exposed children, they included witnessed violence as well as physical, sexual, and emotional abuse, and neglect within the DV category. In summary, in two of the three selected studies, children who were rated as positive for DV were not only exposed to interparental violence, but were also victims of FV.

## 4. Discussion

### 4.1. Does Screening for DV Increase the Opportunity of Detecting Exposed Children?

The current literature review shows prevalence rates of DV among mothers referring to pediatric clinics that fall between 10% and 20%, while previous observations estimate global lifetime prevalence of DV among women to be 30% (slightly lower in high income countries) [3]. In this case, DV screening within pediatric clinical setting seems to not be able to catch all the exposed families.

However, it shows good potential to detect those women who are seeking help [23]. An active plan for DV detection through the use of questionnaires, indeed, significantly increases disclosure rates [21,22], especially regarding past experiences, and women show good acceptance of it (70–80% acceptance rates [12,24,27]). The American Academy of Pediatrics, indeed, strongly recommends pediatric involvement by screening mothers for DV in order to prevent child maltreatment [14].

### 4.2. Where to Screen?

A total of 18 out of 23 studies were set in a pediatric setting, while only 5 were conducted in a CAMHS setting. This evidence is alarming, since children exposed to violence are at high risk of developing emotional and behavioral problems [8], and thus necessitate early referral to CAMHS practitioners. Psychiatrists’ awareness of the topic is even more urgent, since meta-analytic longitudinal studies have shown that the outcome of exposed children is not pre-determined by the event of exposure itself [45] and interventions can lead to healthy adjustment [8]. We found that 48% of studies on DV in the pediatric setting were conducted in primary care centers, while 22% were conducted in EDs. Disclosure rates were not markedly different. Pediatric primary care clinics offer the advantages of time availability and a trustworthy, longitudinal relationship between the healthcare provider and family [26]. Moreover, the lack of punctual child health monitoring itself has been described as a particular aspect of families with potential DV [17]. EDs are the most easily accessible clinical services. They might help to screen women without the violent partner present, which would facilitate DV disclosure [29]. In contrast, the frenetic timing and impossibility of performing screening during night hours or when the conditions of the child are too severe reduce screening opportunities [27]. The study by Cruz et al. highlighted the possibility of disclosure among inpatients [32], which appears to be an interesting and underexplored topic, while no clear advantages have emerged in terms of choosing primary care vs. EDs.

### 4.3. When to Screen: Risk-Based Versus Universal

All the examined studies conducted universal screening, although universal screening for DV within the clinical setting has been criticized [46]. Current WHO guidelines suggest risk-based interviews only and a list of risk factors suggestive of DV is available [47], although they do not allow practical interpretation. A review of the current literature highlighted young maternal age [18], not having completed high school [27], being in a relationship other than a first marriage, having four or more children, being WIC eligible, having previously “no-showed” for a well-child visit [12] and having sole custody or pending legal concerns [35] as risk factors. The strength of this evidence is still very limited, since each factor is reported in only one or a few studies with limited samples. Of the potential risk factors, punctuality in providing healthcare for children [12] and custody status [35] appear to be potentially effective elements for the detection of DV by clinical practitioners and should be better investigated. Lower disclosure rates were found in studies with a high number of patients covered by Medicaid (3.7% to 15%) [21,22], although Siegel et al. suggested that indigent patients more readily disclose recent abuse [20]. No differences in disclosure rates emerged between different environmental and sociodemographic settings (i.e., suburbs, large cities).

Meta-analytic studies are needed to extract quantitative data on risk factors related to DV in the pediatric healthcare setting.

### 4.4. Who to Screen?

Most screening programs for DV are addressed to women [46]. Indeed, most selected studies used screening tools only on mothers, and the presence of a male partner was an exclusion criterion [20,26]. This approach guarantees safety, although future screening programs should include both male and female caregivers [6]. Paternal reports of DV are important to identify bilateral violence and to have more consistent data on children’s problems [34]. Data are scarce regarding caregivers who lost child custody or who were already in contact with social services, and in many studies, they were either excluded or not mentioned. The role of children and adolescents in DV disclosure and its practical implications have been explored in just three of the selected studies [2,36,37]. Ethical questions arise when interviewing young children on this topic [48]; children might provide incoherent narratives or remove events as a means of defense [49] and currently used screening tools were found to have poor psychometric properties [50]. Only one selected study used an oral interview for children [37], consistent with the suggestion that once children can speak, healthcare providers should try direct assessment [6]. The prevalent age group was 12–17 years [2,36,37] among the studies in children, while for children aged 0–7 years, the caregivers were interviewed. Therefore, it appears that maternal screening tools are the most feasible and widely used tools to assess child exposure to DV in the pediatric setting, especially for young children, who are disproportionately represented among families characterized by current DV [4,51,52]. More direct involvement of minors with the appropriate instruments and an empathic approach should not be excluded, especially after a certain age.

### 4.5. How to Screen? Instruments and Cautions

It is necessary to directly ask and involve families in DV screening to obtain a higher disclosure rate [31]. An active plan for DV detection using questionnaires increases disclosure rates significantly [21,22], and even a brief question on DV can be effective as compared to longer screening methods [17]. The most used instruments were the PVS [41] in EDs and the CTS [40] for more detailed knowledge of exposure. Some instruments with low sensitivity could be useful because they might detect only those subjects who are more inclined to benefit from an intervention due to their willingness to disclose and improve their current situation [23]. Most of the analyzed studies (69.5%) used written or computerized tools, which have been proven to be more effective in adult-care settings [30] and more acceptable to women because they are easier and more confidential [53]. These data have only partially been confirmed within the pediatric setting, where direct verbal questioning was preferred [27] or showed higher disclosure rates [29] in some studies, although not in others [54]. More data need to be collected on the topic. Computer inquiries and audiotape interviews [38] about sensitive issues have been shown to yield the highest detection rates [55]. The topic of displaying posters for children is controversial due to contrasting results [28,33]. Use of nongraphic tools (i.e., not directly mentioning words such as hitting, abuse) could make it possible to screen women, even when children are present [23].

## 5. Limitations

We limited our review to the first point of the screening process, without evaluating experiences about the actual change after detection. The lack of long-term benefits has been shown in adult clinical care as one of the main critical points for promoting DV screening [46]. Moreover, further attention should be given to studies that focus on healthcare providers’ point of view in this process.

## 6. Conclusions

The topic of DV and its impact on child development has captured the attention of clinicians within the last few decades, although research is still scarce, especially within the child psychiatric setting [2,34,35,36,37]. Practitioners who encounter minors in their routine practice need increased awareness of this issue. More detailed guidelines can be useful to facilitate and manage disclosure. An active plan for DV identification significantly increases disclosure rates [21,22] and make it possible to detect those women who are prone to change their condition [23]. Conducting risk-based screening is challenging because the currently identified risk factors are too vague [47]. Greater attention to patient sociodemographic data, custody situations and punctuality in child health monitoring might be helpful to detect DV [12,18,35]. No differences were identified in screening effectiveness between primary care centers and Eds, while inpatient screening should be further evaluated. For young children, parental reports [1] are advisable, while for older children, direct involvement with adequate training and instruments should not be excluded. Data on paternal screening are lacking, but they might help to increase disclosure rates [34]. The use of questionnaires is suggested [21,22] and even a brief question on DV can be effective [17].

## 7. Future Perspectives

In conclusion, the findings from our literature review confirm that actively screening for DV within pediatric clinical services cannot be recommended yet, although it appears crucially important for health professionals to be prepared to actively screen for DV if any doubt arises from the clinical history of the child. We propose that clinicians should be ready to further interview families if any risk factors for DV are detected. We call for an expansion of Thackeray’s indications [14] in order to move towards the development of more detailed clinical guidelines regarding the procedures and interventions.

Further investigations should focus on long-term outcomes after disclosure, as well as on the point of view of healthcare providers. Moreover, a further meta-analytic study of the results obtained in the present review would be useful to examine the quantitative relevance of the topic.

## Figures and Tables

**Figure 1 brainsci-12-01235-f001:**
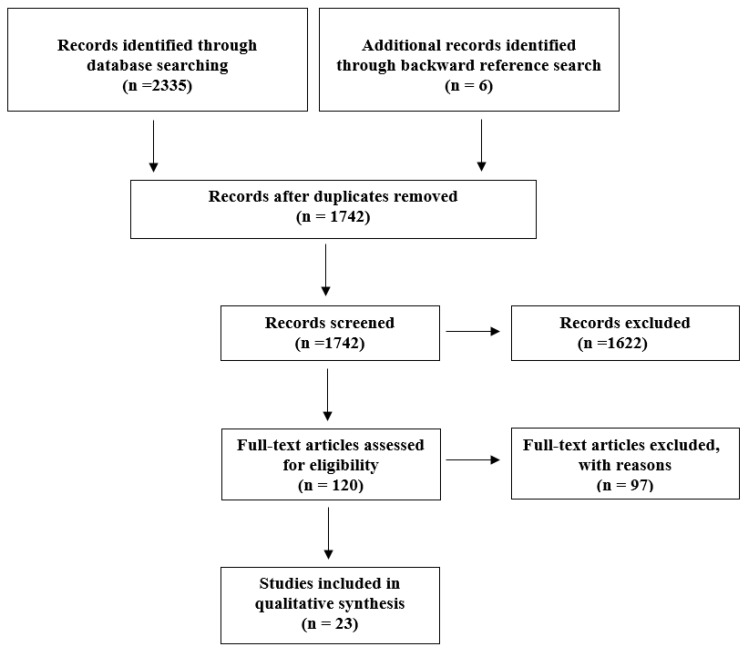
Flowchart for articles selection.

**Table 1 brainsci-12-01235-t001:** Inclusion criteria.

INCLUSION CRITERIA	
Publication-related factors	Peer-reviewed original articles published in English during the time interval January 2000–March 2021
Population	Clinical samples from pediatric clinics or CAMHS
Study designs	Directly involving children and adolescents or their caregivers in the screening process; clearly mentioning domestic violence or intimate partner violence as one of the exposure variables.

CAMHS, children and adolescent mental health clinics.

**Table 2 brainsci-12-01235-t002:** Selected studies, setting, study design.

	Setting	Articles		% (n = 23)
	Pediatric setting			78%
Interviews with mothers	Child care	Almqvist et al., 2018 [24]	12 child healthcare centres in Sweden	48%
		Anderst et al., 2004 [25]	Urban hospital-based pediatric practice in Kentucky	
		Bair-Merritt et al., 2008 [26]	Urban pediatric outpatient clinic	
		Dubowitz et al., 2008 [17]	University-based pediatric primary care clinic	
		Holtrop et al., 2004 [21]	General pediatric clinic at Children’s Hospital of Michigan	
		Klassen et al., 2013 [18]	Urban family medicine residency training clinic in Detroit, Michigan	
		McFarlane et al., 2003 [19]	University of Texas-Houston medical school	
		Parkinson et al., 2001 [12]	Three-paediatrician group practice in Falmouth (Cape Cod)	
		Siegel et al., 2003 [20]	Four sites from the Cincinnati Pediatric Search Group	
		Wahl et al., 2004 [22]	University of Arizona pediatric clinic	
		Zink et al., 2007 [23]	Five pediatric and family medicine practices (75% of the sample was recruited from pediatric practice)	
	Acute care	Bair-Merritt et al., 2006 [30]	Pediatric ED in an urban hospital	22%
		Bair-Merritt et al., 2006 [38]	Urban, academic, tertiary care children’s hospital	
		Newman et al., 2005 [27]	Pediatric ED of Children’s Memorial Hospital in Chicago	
		Randell et al., 2018 [28]	Three pediatric acute care sites within the Midwestern Children’s Hospital system	
		Scribano et al., 2011 [29]	Pediatric ED of the Ohio State University College of Medicine, Columbus, OH	
	Mixed care	Cruz et al., 2013 [32]	Urban tertiary care pediatric hospital	8%
		Kerker et al., 2000 [31]	Pediatric practices in the 13-town Greater New Haven, CT area	
	CAMHS			22%
		Hultmann et al., 2009 [35]	Child and adolescent psychiatric clinic in the Gamlestaden district of Gothenburg	9%
		McDonald et al., 2000 [34]	University of Houston	
Interviews with children		Hultmann et al., 2016 [2]	Child and adolescent psychiatric clinic in the Gamlestaden district of Gothenburg	13%
		Olaya et al., 2010 [36]	Public mental health centres in the metropolitan area of Barcelona	
		Völkl-Kernstock et al., 2016 [37]	Department of Child and Adolescent Psychiatry at the Medical University of Vienna	

CAMHS, children and adolescent mental health clinics; ED, emergency departments.

**Table 3 brainsci-12-01235-t003:** Sociodemographic characteristics of selected samples; DV and IPV. Prevalence rates and risk factors.

Study	N. of Participants	Age and n. of Children	Info about Family	Nationality/Race	Education and Employment/Economic Status	Medical Insurance	Prevalence of DV	Type of DV
Almqvist et al., 2018 [24]	198 mothers		95% biological parents, 97% joint custody. In 39% of mothers, the child was her first child	86% Swedish origin			16% positive for IPV; 13% positive for past abuse (>3 years); 2.5% positive for recent abuse (1–3 years); 0.5% positive for abuse in the past year	15% psychological violence; 10.5% physical abuse; 4% sexual abuse
Anderst et al., 2004 [25]	596 mothers						16% positive for IPV (group 1); 9% positive for IPV in the previous 24 months (group 2); 0% positive for IPV (group 3)	
Bair-Merritt et al., 2006 [30]	269 mothers, mean age: 28.6 years			77% to 79% African American	79% to 85% HS		21% of women in the initial group and 26% in the post-display group (*p* = 0.4)	
Bair-Merritt et al., 2006 [38]	499 mothers; mean age: 32.4–34.0 years			63 to 70% African American	51% to 56% HS; 21% to 25% college		10% of women positive for IPV in the past 12 months	Emotional abuse was most commonly reported
Bair-Merritt et al., 2008 [26]	133 mothers			Predominantly African American		>90% of patients received medical assistance	23% within the last 12 months, out of which 57% stated that at least one child was exposed	
Cruz et al., 2013 [32]	453 exposed mothers; mean age: 25 years	54% ≥2 children	50% lived with their abusive partners	83% Latin or African American			453 referrals over 53 months	
Dubowitz et al., 2008 [17]	198 mothers, 2 fathers; mean age: 25 years	Mean age: 11.8 months	87% single mothers	92% black	Approximately 1/3 < HS; 1/3 HS; 1/3 college; 1/3 employed	93% Medicaid	11.0% positive for physical threats or hurt by partner	
Holtrop et al., 2004 [21]	4084 mothers	Not available		85% black		79% Medicaid; 11% self-pay; 10% private insurance	3.7% in the last 12 months	
Hultmann et al., 2009 [35]	308 mothers	Age range 6–8 years was the most represented	54% sole custody; 32% legal dispute	54% Swedish	56% HS		21% (66/308)	
Kerker et. al., 2000 [31]	939; 65.8% between 31 and 40 years	Not available	73.6% of mothers were married	86.9% white	25% of the sample had a yearly income < 25.000 USD	6.4% food stamps; 7.5% aid to family and dependent children payments	4.2% positive for ever experiencing spousal/partner abuse; pediatricians identified only 0.3% of respondents as experiencing spousal/partner abuse	
Klassen et al., 2013 [18]	121						19.0% of the sample met criteria for DV	
McDonald et al., 2000 [34]	90 dual-parent families; husband mean age: 35.8 years; wife mean age: 33.4 years	Age range 4–7 years		79% Caucasian	Mean USD 33,000		Not available (comparative study abused/non-abused women)	Physical or sexual assaults within the preceding 12 months
McFarlane et al., 2003 [19]	258abused women; 72 non-abused women; age range: 18–44 years			23.2% black; 68.9% Hispanic; 6.7% white; 1.2% Asian	45.2% < USD 10,000 yearly		Husband marital violence was 48% (43/90 families); wife marital violence was 50% (45/90 families)	13 women were kicked, bit, or hit with a fist by their husbands; 7 were beaten up by their husbands; 1 husband used a knife or gun against his wife. 17 husbands were kicked, bit, or hit with a fist; 3 husbands were beaten up
Newman et al., 2005 [27]	451 mothers; mean age: 32 years			Hispanic 42%; black 29%; white 25%	20% <HS; 26% HS; 28% college; 26% completed college. Adjusted % FPL: 10%: <200; 28%: 200–300; 44%: 300–500; 18%: 00		11% positive for IPV in the preceding year	23/50 women experienced physical violence: 19/50 physical assaults; 4/50 sexual assaults, and 4/50 both physical and sexual assaults. 23 women felt unsafe in their current partner relationship; 29 felt unsafe because of a previous partner
Parkinson et al., 2001 [12]	553 mothers; mean age: 32.6 years	Median: 2.0	65.3% married	Not available		63.5% private insurance; 32.5% public insurance (30.5% Medicaid)	14.7% positive for past abuse; 2.5% currently abused; total 16.5%	1/3 reported psychological abuse; 1/3 reported physical hurt; 1/4 reported sexual coercion
Randell et al., 2018 [28]	522 mothers;			55–56% white; 27–31% black; 13–15% Hispanic	55–62% HS or less; 30% unemployed		Predisplay group: 40%; postdisplay: 30%	
Scribano et al., 2011 [29]	13,057 mothers; mean age: 32.6 years		41.3% married; 41.1% single; 8.5% divorced; 4.9% separated; 3.1% unknown; 1.1% widowed		18.2% middle school graduate; 26.9% HS; 28.1% college course but not graduated; 18.6% college graduate; 5.4% professional/graduated; 2.9% unknown	29.6% private/commercial; 58.2% public/Medicaid; 4.6% self-paid; 7.7% Unknown	13.7% among those who used the kiosk	
Siegel et al., 2003 [20]	435 mothers; mean age: 28.6 years	mean age of children: 2.8 years		93% Caucasian		31% Medicaid; 6% self-pay	22% positive for abuse; 16% reported abuse longer than 2 years before the screening; 6% reported abuse within 24 months	
Wahl et al., 2004 [22]	7070 mothers	40% of children aged 1 to 5 years				76% Medicaid; 22% commercial; 2% self-pay	total 15%; 138 (2%) currently abused; 915 (13%) positive for past abuse	
Zink et al., 2007 [23]	393 mothers; mean age: 31 years	Median: 2 years (range 1–9 years)	81.3% married; 13.2% single; 5.5% separated	49.2% white; 50.8% African American	60% > 12th grade; income/year: 34.4% < USD 20,000; 34.4% USD 20,000–40,000; 31.3% USD 40,000		11.2%	

IPV, intimate partner violence; HS, high school; FPL, federal poverty level.

**Table 4 brainsci-12-01235-t004:** Questionnaires, type of questions, administration, acceptability rate of the screening tests.

	Questionnaires Completed by the Mothers	Type of Questions	Method of Administration	Acceptability Rate
Almqvist et al., 2018 [24]	ViF questionnaire; CTS-brief; telephone interview with mothers	(1) Have you (as an adult) been hit, kicked, punched or otherwise hurt by someone? If so, by whom? (2) Have you (as an adult) been ridiculed, threatened, harassed or otherwise hurt by someone? If so, by whom? (3) Do you feel safe in your current relationship? and (4) is there a partner from a previous relationship who is making you feel unsafe now?	Written	71% positive or very positive, 24% neutral, 5% doubtful
Anderst et al., 2004 [25]	Oral survey (group 1 and 2); self-administered 72-item general questionnaire (group 3)	(1) Are you in a relationship now or have you ever been in a relationship in which you have been harmed or felt afraid of your partner? (2) Are you afraid of your current partner? (3) Are you or your child being hurt, hit, or frightened by anyone in your house?	Oral (groups 1 and 2); written (group 3)	-
Bair-Merritt et al., 2006 [30]	Survey questions with five responses on a Likert scale (definitely yes, probably yes, not sure, probably no, and definitely no)	Among others: “I am familiar with the problems of domestic violence because of: [check all that apply: the experience of a friend or relative, personal experience, reading about it, hearing about it on the radio or TV, don’t know or don’t remember, other]”.	Oral	The majority of women in both groups were satisfied with using the method, they would be willing to use the method again and the method was considered a safe way for women to disclose IPV.
Bair-Merritt et al., 2006 [38]	Six general safety questions about fire safety and poisoning prevention and four IPV questions, three from the PVS and an additional question on emotional abuse; dichotomous yes or no answers	Among others: (1) Do you feel safe in your current relationship with your partner? (2) Have you been hit, kicked, punched, or otherwise hurt by a partner within the past year? (3) Is there a partner from a previous relationship who is making you feel unsafe now? Within the past year, has a partner repeatedly used words, yelled, or screamed at you in a way that frightened you, threatened you, put you down, or made you feel rejected?	Written or audiotape	Women preferred the audiotape method, which they considered to be safer and more confidential; women were willing to use the screening method again; women preferred to avoid direct screening.
Bair-Merritt et al., 2008 [26]	Ten-item WEB scale, eight-item CTS-1, questions about children’s domestic violence	Not available	Written	-
Cruz et al., 2013 [32]	Oral questions with phrases from the Institute for Safe Families Pediatric RADAR cards	Example: Because violence is so common in the lives of women, I have begun to ask all of my patients about it. Is there anyone who has physically or sexually hurt you?	Oral	-
Dubowitz et al., 2008 [17]	PSQ in the clinic. PSQ and CTS-2 through computer-assisted self-interview	(1) Have you ever been in a relationship in which you were physically hurt or threatened by a partner? (2) In the past year, have you been afraid of a partner? (3) In the past year, have you thought of getting a court order for protection?	Computer	-
Holtrop et al., 2004 [21]	PVS	(1) Have you been hit, kicked, punched, or otherwise hurt by someone within the past year? If so, by whom? (2) Do you feel safe in your current relationship? (3) Is there a partner from a past relationship that is making you feel unsafe right now? One question modified from the Abuse Assessment Screen: (4) Have you had unwanted or forced sexual contact with someone within the past year?	Written	-
Hultmann et al., 2009 [35]	Questions by the healthcare provider	Not available	Oral	-
Kerker et al., 2000 [31]	Provider Assessment Questionnaire, in particular the section with questions to the mother; checklist for pediatricians	“Have you ever been badly beaten or bruised by another person?” and subsequent queries as to the relationship the woman had with her abuser	Oral	-
Klassen et al., 2013 [18]	CTS-2	Not available	Written	-
McFarlane et al., 2003 [19]	CBCL	Not available	Oral	-
McDonald et al., 2000 [34]	CTS;Short Marital Adjustment Test; CBCL; PC-CTS	Not available	Written	-
Newman et al., 2005 [27]	PVS	(1) Have you been hit, kicked, punched, or otherwise hurt by someone within the past year? If so, by whom? (2) Do you feel safe in your current relationship? (3) Is there a partner from a past relationship that is making you feel unsafe right now?	Written	75% agree
Parkinson et al., 2001 [12]	Written questions	(1) In your current relationship, have you ever been harmed or felt afraid of your partner? (2) In a previous relationship, have you ever been harmed or felt afraid of your partner? (3) Has your current or past partner harmed any of your children? (4) Are there any guns in your house?	Written	82.8% favored being asked about IPV, 12.1% were neutral, and 5.1% opposed. Response to this question did not differ for those who had/did not have a history of MDV
Randell et al., 2018 [28]	Computer-assisted self-interview on IPV in response to healthcare provider screening using a 3-point Likert scale	Not available	Computer	More subjects in the postdisplay group approved of the display of IPV materials in pediatric emergency department/urgent care center restrooms (94% pre vs. 98% post, *p* = 0.04) and examination rooms (94% pre vs. 98% post, *p* = 0.01)
Scribano et al., 2011 [29]	Computerized screening kiosks using “home safety screening’’, adapted by Bair-Merrit et al., 2006, and the PVS, adding questions on emotional and sexual abuse	(1) Have you been hit, kicked, punched, or otherwise hurt by someone within the past year? (2) Do you feel safe in your current relationship? (3) Is a partner from a previous relationship making you feel unsafe now? (4) Within the past year, has a partner repeatedly used words, yelled, or screamed at you in a way that frightned you, threatened you, put you down, or made you feel rejected? (5) Have you had unwanted, forced sexual contact with someone in the past year?	Computer	-
Siegel et al., 2003 [20]	Oral questions	(1) Are you in a relationship now or have you ever been in a relationship in which you have been harmed or felt afraid of your partner? (2) Has your partner ever hurt any of your children? (3) Are you afraid of your current partner? (4) Do you have any pets in the house? (5) Has your partner or child ever threatened or hurt any of the pets? (6) Are there any guns in your house?	Oral	-
Wahl et al., 2004 [22]	Child Safety Questionnaire	(1) Have you ever been in a relationship with someone who has hit you, kicked you, slapped you, punched you, or threatened to hurt you? (2) What about your current relation? (3) When you were pregnant did anyone ever physically hurt you? (4) Are you in a relationship with someone who yells at you, calls you names, or puts you down?	Written	-
Zink et al., 2007 [23]	Five DV screening questions CTS-2	(1) How do you and your partner work out arguments? (2) In general, how do you describe your relationship? (3) How is your partner treating you and the children? (4) Do you feel safe in your current relationship? (5) Considering your current partners or friends or any past partners or friends, is there anyone who is making you feel unsafe now?	Questions in three randomly selected formats (written, verbal, palmtop computer) CTS written	-

CBCL, Child Behavior Checklist; CTS, Conflict Tactics Scale; DV: domestic violence; IPV; intimate partner violence; PC-CTS: Parent-Child Conflict Tactics Scale; PSQ, Parent Screening Questionnaire; PVS, Partner Violence Screen; ViF, violence in the Family; WEB, Women’s Experience with Battering scale.

**Table 5 brainsci-12-01235-t005:** Percentage of usage of different screening tools.

Screening Instruments	Selected Articles	% of Screening Instrument Usage in All Studies on Maternal Screening for DV
Violence in the Family (ViF) questionnaire	Almqvist et al., 2018 [24]	5%
Conflict Tactics Scale (CTS)	Almqvist et al., 2018 [25 (CTS-B); Bair-Merritt, 2008 (CTS-1 [26]); Dubowitz et al., 2008 [17] (CTS-2); Klassen et al., 2013 [18] (CTS-2); McDonald et al., 2000 [34] (CTS-1, PC-CTS); Zink et al., 2007 [39] (CTS-2)	30%
Women’s Experience with Battering scale (WEB scale)	Bair-Merritt, 2008 [26]	5%
Parent Screening Questionnaire (PSQ)	Dubowitz et al., 2008 [17]	5%
Partner Violence Screen (PVS)	Holtrop et al., 2004 [21]; Newman et al., 2005 [27]; Bair-Merritt et al., 2006 [38]	15%
Short Marital Adjustment Test	McDonald et al., 2000 [34]	5%
Child Safety questionnaire	Wahl et al., 2004 [22]	5%
Safety questionnaire	Bair-Merrit et al., 2006; Scribano et al., 2011 [29] (modified)	10%
Other oral interviews	Almqvist et al., 2018 [24]; Anderst et al., 2004 [25]; Bair-Merritt, 2006 [30]; Cruz et al., 2013 [32]; Hutleman et al., 2009 [35]; Kerker et al., 2000 [31]; Zink et al., 2007 [39]	35%
Other written questionnaires	Bair-Merritt, 2008 [26]; Parkinson et al., 2001 [12]; Zink et al., 2007 [39]	15%
Other computer questionnaires	Randell et al., 2018 [28]; Zink et al., 2007 [39]	5%

**Table 6 brainsci-12-01235-t006:** Studies on children.

	Hultman, 2016 [2]	Völkl-Kernstock, 2016 [37]	Olaya, 2010 [36]
Sample type	Outpatients	Outpatients	Patients
Total sample size	305 responders	946	520
Age span and average in total sample	Age span: 9–17 years (69% were 13–17 years old)	Age span: 6–20 years	Age span: 8–17 years Mean age = 13.2 years (SD = 2.5)
Gender	M = 50.8%	-	M = 54.3%
Sample exposed to domestic violence	146 (48% of the total sample, 71.2% out of the 205 positive for any form of violence exposure)	257 (27% of the total sample, 35.5% out of the 723 positive for any form of violence exposure)	100 children (19.2% of the total sample)
Age span and mean age of exposed children	Age span: 12–17 years was the most represented	Age span: 11–15 years was the most represented	Mean age = 13.5 (SD = 2.6)
Gender distribution in exposed children	F = 82 (56%) M = 64 (44%)	F = 134 (52%) M = 123 (48%)	F= 55.3% M = 44.7%
Instruments to assess violence exposure	Life Incidence of Traumatic Events (LITE) “Have you seen parents hitting each other or destroying furniture?” Child abuse was investigated through the following items: “being beaten at home”, “being tied up or locked up at home”, and “being subjected to sexual abuse at home”. 1 to 3 scale: not at all, a little, a lot	Childhood Trauma Interview (CTI) Brief assessment of multiple dimensions of the following six types of childhood interpersonal trauma: neglect, social violence, emotional abuse or assault, physical abuse or assault, sexual abuse or assault, witnessing violence	Children’s perception of interparental conflict scale “Have you ever seen your parents pushing or shoving each other when they quarrel?”; “Have you ever seen your parents hitting each other during an argument?” or “Have you ever seen your parents breaking or throwing objects during an argument?” (3-point Likert-type scale)
Instruments to evaluate psychopathology	- Strengths and Difficulties Questionnaire (SDQ) - Clinician-rated Global Assessment of Functioning (GAF)	Not specified	- Diagnostic Interview for Children and Adolescents (DICA-IV) - Child Behavior Checklist (CBCL) - Child and Adolescent Functioning Assessment Scale (CAFAS) - AC-Self-Concept Questionnaire - Risk factors schedule
Instruments to evaluate parental factor	-	-	- Scale for parental style (EMBU) - SCL-90-R
Socio-environmental factors	Exposed patients were more likely to live with none of their parents or under one-parent custody than children in the no violence group and were more often born abroad.	-	Higher frequency of one-parent families; more common economic problems in exposed families.

## Data Availability

No new data were created or analyzed in this study. Data sharing is not applicable to this article.

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
