# Peer review of "Feasibility of Screening Programs for Domestic Violence in Pediatric and Child and Adolescent Mental Health Services: A Literature Review"

_brainsci, 2022, doi:10.3390/brainsci12091235_

Round 1
Reviewer 1 Report
This study is considered a valuable research topic by systematically evaluating domestic violence in pediatric care.
This study focuses on the collection of domestic violence (DV) studies. Nonetheless, the Introduction and Materials and Methods descriptions do not provide a more direct definition of DV from researchers. Hopefully, the researchers can remove the vague descriptions from the text and make it more clearly defined.
In addition, this study includes studies published in English between January 2000 and March 2021 on children under 18 years of age or their caregivers in pediatric or mental health clinics investigating screening strategies for DV exposure; the researchers need to clarify the following.
1. whether the definition of domestic violence (DV) programs is consistent across countries
2. Why was January 2000 - March 2021 chosen as the time interval for this study? What does it mean? Does it have any special meaning?
3. Is any relevant research or international policy between January 2000 and March 2021 that further clarifies or describes DV?
In addition to analyzing the literature, it is hoped that the researcher will draw more constructive and applied findings based on the data collected, which could add value to this study.
Reviewer 2 Report
The main characteristic of the present review is a systematic approach to the literature search. But this does not characterize the review itself as a systematic review. According to PRISMA guidelines, the review should have a meta-analytic data investigation. Also, studies that are not possible do not analytic searches such as case reports, letters, or correspondences should not be added to systematic reviews. Therefore, the review information should be changed to a literature review instead of a systematic review to avoid future issues.
Page MJ, et al. Updating guidance for reporting systematic reviews: development of the PRISMA 2020 statement. J Clin Epidemiol. 2021 Jun; 134:103-112. DOI: 10.1016/j.jclinepi.2021.02.003. Epub 2021 Feb 9. PMID: 33577987.
1. The title should be changed according to a literature review. Also, the current title is not associated with the main findings or investigations of the manuscript. It is advised to change to avoid misunderstandings.
2. Abstract
A clear description of the methodology should be provided, such as databases searched and years.
Results should present specific data descriptions like percentages.
What are future studies provided? One of the main concerns regarding a literature review with a systematic approach is reporting fails in the reporting system. Could the authors provide some sentences about it?
3. Introduction
The present introduction does not provide substantial background for the study. Could the authors provide more details about what this study is needed? It is advised at least two more paragraphs concerning the study's main idea.
The objective should not have a subheading.
Removing any information regarding PRISMA guidelines is advised to avoid future issues.
4. Methods
It is advised to provide a complete description of every term at first appearance. E.g., "APA"
Could the authors provide examples of disagreements between the authors during the studies?
Do the authors have access to every one of the articles that they searched? Do they email other authors requesting manuscripts?
5. Future directions
A specific chapter about future directions should be provided.
Round 2
Reviewer 2 Report
Satisfactory.